# Kernel Neural Operators (KNOs) for Scalable, Memory-efficient, Geometrically-flexible Operator Learning

## Abstract

This paper introduces the Kernel Neural Operator (KNO), a novel operator learning technique that uses deep kernel-based integral operators in conjunction with quadrature for function-space approximation of operators (maps from functions to functions). KNOs use parameterized, closed-form, finitely-smooth, and compactly-supported kernels with trainable sparsity parameters within the integral operators to significantly reduce the number of parameters that must be learned relative to existing neural operators. Moreover, the use of quadrature for numerical integration endows the KNO with geometric flexibility that enables operator learning on irregular geometries. Numerical results demonstrate that on existing benchmarks the training and test accuracy of KNOs is higher than popular operator learning techniques while using at least an order of magnitude fewer trainable parameters. KNOs thus represent a new paradigm of low-memory, geometrically-flexible, deep operator learning, while retaining the implementation simplicity and transparency of traditional kernel methods from both scientific computing and machine learning.

## 1 Introduction

Operator learning is a rapidly evolving field that focuses on the approximation of mathematical operators, often those arising from partial differential equations (PDEs). Modern approaches leverage machine learning (ML) to approximate complex operator mappings between infinite-dimensional spaces. Recent approaches include the DeepONet family of neural operators Lu et al. (2021; 2022); Zhang et al. (2023); Jin et al. (2022), the family of Fourier neural operators (FNOs) Li et al. (2021); Kovachki et al. (2021); Li et al. (2023; 2024), graph neural operators (GNOs) Li, Zongyi and Kovachki, Nikola and Azizzadenesheli, Kamyar and Liu, Burigede and Bhattacharya, Kaushik and Stuart, Andrew and Anandkumar, Anima (2020); Li et al. (2020), and kernel/Gaussian-process-based methods Batlle et al. (2024).

In this paper we propose a new method, **the kernel neural operator (KNO)**, that improves upon existing operator learning techniques (namely FNOs and GNOs) by leveraging kernel-based deep integral operators. While numerous works have shown that such methods can produce accurate approximations of non-linear operators, e.g. Li et al. (2021); Kovachki et al. (2021), this accuracy comes at the cost of an extremely large model parameterization that induces onerous memory and training requirements. These challenges arise because existing methods choose specific discretizations of the aforementioned integral operators without directly learning the kernels; for example, the FNO uses a fast Fourier transform on an equispaced grid to learn the kernel in spectral space while the GNO uses a graph parametrization to discretize the integral. This implicit kernel learning also prevents some desirable properties from being directly encoded into the kernel and enforces other properties that may not be necessary: e.g., FNOs implicitly restrict the class of learnable kernels to radial and periodic ones.

In contrast to existing approaches, the KNO directly uses closed-form trainable kernels in conjunction with quadrature to approximate the action of its integral operators. Our numerical results show that the ability to utilize specific types of trainable kernels – namely sparse and compactly-supported kernels – significantly improves the accuracy of the operators learned. Moreover, our approach comes with many other immediate benefits: (1) the use of quadrature allows us to tackle operator learning on irregular domains with little to no difficulty; (2) the use of specific closed-form

trainable kernels allows us explicit control over the number of trainable parameters; (3) the use of these explicit kernels allows us to directly operate on point-cloud inputs rather than being tied to a regular grid; and (4) the use of closed-form trainable kernels improves the transparency of our neural operator architecture. Additionally, like the FNO family of neural operators, the KNO is formulated entirely in **function space** and therefore inherits the associated benefits: e.g., zero-shot super resolution, superior generalization capabilities in both input and output spaces, discretization-invariance in the input domain, and the ability to evaluate the architecture at arbitrary locations in the domain of the learned operator.

In addition to the beneficial properties of the KNO outlined above, KNOs obtained **state-of-the-art accuracy** on a variety of challenging operator learning benchmark problems involving PDEs, including those on non-rectangular domains. Moreover, the KNO was able to accomplish this with **1-2 orders of magnitude fewer trainable parameters** than reported in the literature for other neural operators.

## 1.1 CONNECTIONS TO OTHER METHODS

Other operator learning techniques can handle irregular domains but possess restrictions. For example, the DeepONet family of architectures Lu et al. (2022); Peyvan et al. (2024) can handle input and output functions sampled on irregular domains, but require that all input functions must be sampled at the same input domain locations. The FNO was generalized to tackle arbitrary domains as well, first through the "dgFNO+" architecture Lu et al. (2022), then more recently through the geoFNO architecture Li et al. (2023; 2024). The latter accomplished this by simultaneously learning both the operator and a mapping from input locations to a regular grid, allowing for the use of the FFT. However, such mappings may not always exist or be feasible to compute. In contrast, the KNO possesses none of these limitations, requiring only information transfer to a set of quadrature points through straightforward function sampling, in a manner similar to Solodskikh et al. (2023) (though the latter as presented was restricted to regular grids). In summary, the KNO leverages the rich literature on compactly-supported kernels and the even richer literature on quadrature, resulting in a relatively simple, parsimonious, and powerful architecture.

More broadly, kernel methods have been in use for decades in machine learning Rasmussen & Williams (2006); Cortes & Vapnik (1995); Boser et al. (1992); Broomhead & Lowe (1988); Sharma & Shankar (2022). Kernels have also been *designed* to fit data McCourt et al. (2018); Fasshauer & McCourt (2015) and sparsified using partition-of-unity approximation Han et al. (2023). Additionally, kernel methods based on regression have been applied recently to operator learning problems Batlle et al. (2024) using an extremely small number of trainable parameters, albeit with generally lower accuracy than the KNO. The KNO falls on the spectrum between these kernel/GP operator learning methods and FNOs (which are also kernel-based), being more parameterized than the former and less than the latter. Kernels have also been heavily leveraged in scientific computing within (shallow) integral operators Gingold & Monaghan (1977); Peskin (2002); Kassen et al. (2022a;b); Hsiao & Wendland (2008); Cortez (2001); Shankar & Olson (2015) or as generators of finite difference methods Wright & Fornberg (2006); Fornberg & Flyer (2015); Bayona et al. (2019); Fasshauer & McCourt (2015); Shankar et al. (2014); Shankar & Fogelson (2018), and more recently to accelerate the training of physics-informed neural networks Sharma & Shankar (2022). Our development of the KNO was the result of aggregating insights from this very broad body of work on kernel methods and applying them deep learning and, more specifically, deep operator learning.

**Limitations**: Much like the FNO and other neural operators, our method is subject to a curse of dimensionality, in our case for two reasons: first, because the kernel interpolant in our pipeline requires decreasing fill distance of the data sample locations in order to converge; and second, because the number of quadrature points in most standard quadrature rules grows exponentially with dimension (the FNO faces the same problem). There are some well-known approaches to ameliorate these issues Zech & Schwab (2020); L'Ecuyer (2018), but we opt for a general presentation and so do not use those approaches here. Finally, our results for other methods were based on reported data from Lu et al. (2022); Batlle et al. (2024), not our own implementations; reported parameter counts for those methods may hence not be optimal.

## 2 KERNEL NEURAL OPERATORS (KNOs)

Given Euclidean domains $\Omega_u, \Omega_y$ and $d_u, d_y \in \mathbb{N}$, neural operators learn mappings from a Banach space $\mathcal{U} = (\Omega_u; \mathbb{R}^{d_u})$ of $\mathbb{R}^{d_u}$-valued functions to a Banach space $\mathcal{Y} = \mathcal{Y}(\Omega_y; \mathbb{R}^{d_y})$ of $\mathbb{R}^{d_y}$- valued functions through supervised training on a finite number of input-output measurements. From a statistical learning point of view, neural operators are learned from measurements of input functions drawn from a probability measure $\nu$ on $\mathcal{U}(\Omega_u; \mathbb{R}^{d_u})$. In the following, we present the formulation of KNOs, which are a special class of neural operators that leverage properties of certain kernel functions for the benefit of efficiency and accuracy.

### 2.1 FUNCTION SPACE FORMULATION

Let $\mathcal{G}$ be an unknown operator we wish to learn that is an element of the $L^2$-type Bochner space $L_\nu^2(\mathcal{U}; \mathcal{Y})$, i.e., $\mathcal{G}$ is a mapping from $\mathcal{U}$ to $\mathcal{Y}$ that is Borel-measurable with respect to the probability measure $\nu$ on $\mathcal{U}$. We are interested in learning a KNO $\mathcal{G}^\dagger$ that minimizes a loss function $L$ measuring how well functions predicted by the operator match the training data. For example, the loss function may be the $L_\nu^2$ norm on operators,

$$L(\mathcal{H}, \mathcal{G}) = \|\mathcal{H} - \mathcal{G}\|_{L_\nu^2(\mathcal{U};\mathcal{Y})}^2 = \mathbb{E}_{f \sim \nu} \|\mathcal{H}(f) - \mathcal{G}(f)\|_{\mathcal{Y}}^2,$$

which is the loss function we use in our experiments, with the addition of some regularization on the kernel scale parameters and a scaling term to account for relative error. The corresponding statistical learning problem is

$$\mathcal{G}^\dagger = \underset{\mathcal{H} \in \text{KNOs}}{\arg\min} L(\mathcal{H}, \mathcal{G}), \tag{1}$$

where KNOs are operators of the form

$$\mathcal{H} = \mathcal{P} \circ \sigma \circ \mathcal{I}_L \circ \sigma \circ \mathcal{I}_{L-1} \circ \sigma \circ \ldots \sigma \circ \mathcal{I}_1 \circ \mathcal{L}. \tag{2}$$

The operators $\mathcal{I}_\ell, \mathcal{L}, \mathcal{P}$ are all trainable, and an appropriate parameterization of these defines a KNO. The function $\sigma$ is a nonlinear activation that operates pointwise: $(\sigma \cdot f)(x) := \sigma(f(x))$. Additionally, the initial operator $\mathcal{L}$ is a *lifting operator* that takes $\mathbb{R}^{d_u}$-valued functions to $\mathbb{R}^{p_0}$-valued functions, where $p_0 \in \mathbb{N}$. The ultimate operator $\mathcal{P}$ is a *projection operator* that takes $\mathbb{R}^{p_L}$-valued functions and compresses them down to $\mathbb{R}^{d_y}$-valued functions. The dimensions $p_0, \ldots, p_L$ denote the number of *channels* in the architecture.

The workhorses of the KNO, containing most of the novelty and impact, are the latent operators $\mathcal{I}_\ell$, which are linear operator mappings from vector-valued functions to vector-valued functions. These operators are defined by,

$$\mathcal{I}_\ell(\boldsymbol{f}_\ell) = \int_{\Omega_{\ell-1}} \boldsymbol{K}^{(\ell)}(x,y)\boldsymbol{f}_\ell(y)dy, \quad \boldsymbol{f}_\ell : \Omega_{\ell-1} \to \mathbb{R}^{p_{\ell-1}}, \quad \boldsymbol{g}_\ell = \mathcal{I}_\ell(\boldsymbol{f}) : \Omega_\ell \to \mathbb{R}^{p_\ell},$$

where $\boldsymbol{K}^{(\ell)} : \Omega_\ell \times \Omega_{\ell-1} \to \mathbb{R}^{p_\ell \times p_{\ell-1}}$ is a matrix-valued kernel function,

$$\boldsymbol{K}^{(\ell)}(x,y) = \begin{pmatrix} K_{1,1}^{(\ell)}(x,y) & K_{1,2}^{(\ell)}(x,y) & \cdots & K_{1,p_{\ell-1}}^{(\ell)}(x,y) \\ K_{2,1}^{(\ell)}(x,y) & K_{2,2}^{(\ell)}(x,y) & \cdots & K_{2,p_{\ell-1}}^{(\ell)}(x,y) \\ \vdots & \vdots & \ddots & \vdots \\ K_{p_\ell,1}^{(\ell)}(x,y) & K_{p_\ell,2}^{(\ell)}(x,y) & \cdots & K_{p_\ell,p_{\ell-1}}^{(\ell)}(x,y) \end{pmatrix} \in \mathbb{R}^{p_\ell \times p_{\ell-1}}, \tag{3}$$

$p_\ell$ is the dimension of the range of the function that is output from $\mathcal{I}_\ell$, and $\Omega_l$ is its domain. In contrast to the FNO family of neural operators, the KNO *directly discretizes the integral operators $\mathcal{I}$ using quadrature and closed-form trainable kernels*. Further, we determined that the KNO obtained the best accuracy when $\boldsymbol{K}^{(\ell)}$ was chosen from the class of $2k$-smooth compactly-supported positive-definite functions: *i.e.*, $\boldsymbol{K}^{(\ell)} \in C_c^{2k}(\Omega_\ell \times \Omega_{\ell-1}; \mathbb{R}^{p_\ell \times p_{\ell-1}})$. However, at a specific stage in our pipeline, we also leverage a kernel with infinite smoothness. These choices simultaneously provided model capacity and computational efficiency. We now describe the KNO in further detail; a block diagram is shown in Figure 1, while mathematical formulations are shown in (2) and (12).

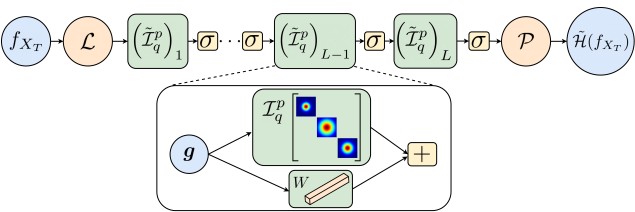

Figure 1: A schematic of the KNO as defined by (12).

**Integral operators**   FNOs use an implicitly-defined, dense, matrix-valued kernel that couples all channels of the architecture. In contrast, the KNO enforces sparsity on this global matrix by utilizing a diagonal matrix-valued kernel. While we briefly experimented with other choices such as tridiagonal matrix-valued kernels (see Appendix A.3), we found that the diagonal kernel resulted in the fewest trainable parameters without degrading accuracy. Choosing a diagonal matrix-valued kernel amounts to making the simple choices of (i) $p_0 = p_1 = \cdots = p_{L-1} = p$ and (ii) choosing $\boldsymbol{K}^{(\ell)}$ as a diagonal kernel. This has the effect of creating $p$ *channels*. The diagonal elements of $\boldsymbol{K}^{(\ell)}$ are further compressed by making only $q \leq p$ of them trainable, resulting in $q$ trainable kernel parameters per index layer $\ell$. We also choose $\Omega_\ell = \Omega \subset \mathbb{R}^d$ for all $\ell \in [L]$ so that we may use radial kernels. In particular, for $\ell \in [L-1]$, the function $K_i^{(\ell)}$ for each $i \in [q]$ is chosen as $K_i^{(\ell)}(x,y) = \phi_{\ell,i}(\|x-y\|)$, where $\phi_{\ell,i} : \mathbb{R} \to \mathbb{R}$ is a radial kernel function with a trainable compact support parameter $\epsilon_{\ell,i}$ to allow flexibility in sparsity; we explicitly provide our choice of $\phi$ in (4), and the final layer $\ell = L$ is described later. We choose $q$ independently of $\ell$, so that these integral operators amount to $(L-1)q$ trainable parameters. Notationally, we will refer to our particular parameterization of the general kernel $\mathcal{I}_\ell$ as $(\mathcal{I}_q^p)_\ell$:

$$\left(\mathcal{I}_q^p\right)_\ell (\boldsymbol{f}) = \int_\Omega \boldsymbol{K}^{(\ell)}(x,y)\boldsymbol{f}(y)dy \qquad\qquad \boldsymbol{K}^{(\ell)} \text{ as in (3).}$$

As in many neural operator formulations, we augment these kernel operations at the discrete level with dense cross-channel affine transformations ("pointwise convolutions") having trainable parameters. We describe this later when we introduce our discretization of the latent space.

## 2.2 Choosing kernels

Each layer of the KNO contains a set of kernels. In this paper, for all but the last layer, we used compactly-supported radial kernels of the Wendland type. The Wendland kernels are a family of **compactly-supported, positive-definite** kernels with smoothness class $s$ (up to some finite dimension $d$), and have been used extensively in scientific computing applications Wendland (2005); Schaback & Wendland (2006); Fasshauer (2007); more recently, Wendland kernels have also been used in machine learning applications Han et al. (2023). The use of Wendland kernels results in a parsimonious parameterization of the KNO, improved training characteristics, spatial sparsity for computational efficiency, and superior accuracy over other choices. Specifically, we used the $C^4\left(\mathbb{R}^d\right)$ compactly-support radial and isotropic Wendland kernel Wendland (1995; 1998):

$$\phi_\epsilon(r) = \left(\text{ReLU}\left(1 - \epsilon r\right)\right)^6 \left(35(\epsilon r)^2 + 18(\epsilon r) + 3\right), \tag{4}$$

where $\epsilon \in \mathbb{R}^+$ is the sole trainable parameter, and $d \leq 3$. The parameter $\epsilon$ serves to both control the flatness of $\phi$ and its region of compact-support: the radius of support $\rho$ is given by $\rho = \frac{1}{\epsilon}$. Since $\phi$ is compactly-supported, a matrix of evaluations of $\phi$ is sparse.

While Wendland kernels can theoretically be used for all layers of a KNO, we found that using an expressive globally-supported kernel within the final integral operator resulted in the best accuracy over a wide range of problems. Specifically, for the last layer we used a spectral mixture kernel constructed as a trainable mixture of two Gaussians Wilson & Adams (2013): for $\boldsymbol{K}^{(L)}$ as in (3), we defined

$$K_i^{(L)}(x,y) = \psi(x-y), \quad \psi(\tau) = \sum_{r=1}^{2} \lambda_r \prod_{p=1}^{d} \cos\left(2\pi\tau_p \mu_r^{(p)}\right) e^{-2\pi^2 \tau_p^2 \nu_r^{(p)}}, \tag{5}$$

where $\tau_p$ is the $p$-th component of $\tau$, and each Gaussian $r = 1, 2$ has a trainable parameter $\boldsymbol{\mu}_r \in \mathbb{R}^d$ and trainable covariances (shape parameters) $\nu_r^{(1)}, \ldots, \nu_r^{(d_y)}$. As with the other layers, we use latent kernels to form the diagonal of the matrix-valued kernel such that the kernel $K_i^{(L)}$ has different trainable parameters from $K_j^{(L)}$ for $i \neq j$.

**Why these kernels?** Unlike existing methods, such as the FNO, the class of kernels used by a KNO can be finely controlled. We leveraged this fine control and investigated compactness of the spectrum of the neural tangent kernel (NTK) matrix of the KNO for different kernel choices. We then chose the KNO architecture whose NTK spectrum indicated the greatest robustness to hyperparameter choices. See Appendix A.5 for details.

## 2.3 SAMPLING AND OUTER DISCRETIZATION

Numerically constructing (2) requires sampling from $\nu$ and a discretization of $\| \cdot \|_{\mathcal{Y}}$. To this end, we trained our KNOs using $M$ independent and identically distributed input samples of functions $f^{(m)} \sim \nu$ drawn from $\mathcal{U}$ and the associated output function data $g^{(m)} := \mathcal{G}(f^{(m)})$, for $m \in [M]$. We used a *training grid*, $X_T = \{x_j\}_{j \in [N_T]} \subset \Omega$, to both discretize the input and output functions $f^{(m)}$ and $g^{(m)}$ and to approximate the norm $\| \cdot \|_{\mathcal{Y}}$. Hence, during learning we optimized

$$\|\mathcal{H} - \mathcal{G}\|^2_{L^2_\mu(\mathcal{U}, \mathcal{Y})} \overset{f^{(m)} \sim \nu}{\simeq} \frac{1}{MN_T} \sum_{(m,j) \in [M] \times [N_T]} \left\| \mathcal{H}(f_{X_T}^{(m)})(x_j) - g^{(m)}(x_j) \right\|_2^2. \tag{6}$$

The input function $f_{X_T}$ is defined as a (trainable) kernel interpolant on the training grid:

$$f_{X_T} = \sum_{n \in [N_T]} c_n K(x, x_n), \tag{7}$$

where the $c_n$ are determined through a size-$N_T$ linear system solve that enforces $f_{X_T}(x_n) = f(x_n)$. This interpolant allows for evaluation of $f$ off of the training points $X_T$, and in particular, at the quadrature points to be introduced shortly. We chose the kernel as $K(x, y) = \phi(\|x - y\|)$ from (4), which ensured that the linear system was sparse and well-conditioned. We emphasize that our choice to evaluate the outputs of $\mathcal{H}$ at $X_T$ was only to enable simple training of our KNOs; for generalization and super-resolution, one can evaluate the output of $\mathcal{H}$ on any desired grid.

## 2.4 LATENT SPACE DISCRETIZATION: QUADRATURE ON GENERAL DOMAINS

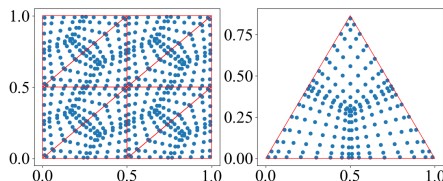

Figure 2: Clustered quadrature points on $[0,1]^2$ (left) and a reference triangle (right).

In order to propagate $f_{X_T}$ through $\mathcal{H}$ in (6), one must discretize all the integral operators; we accomplished this with quadrature. This first requires that we evaluate the kernel interpolant (7) at some set of quadrature points (described further below). This KNO methodology of directly discretizing the integrals via quadrature is a crucial difference compared to other neural operator approaches. Consider the discretization of an integral operator $\int_\Omega K(x, y) f(y) d\mu(y)$ that acts on a scalar-valued function $f : \mathbb{R}^d \to \mathbb{R}$; the generalization to vector-valued functions is straightforward. Then given a *quadrature rule* $\{w_i^q, y_i^q\}_{i=1}^{N_Q}$, where $w_i^q \in \mathbb{R}$ are *quadrature weights* and $y_i^q \in \mathbb{R}^d$ are *quadrature points*, the quadrature-based discretization of a KNO integral operator is

$$\int_\Omega K(x, y) f(y) d\mu(y) \approx \sum_{i=1}^{N_Q} w_i^q K(x, y_i^q) f(y_i^q). \tag{8}$$

In general, the choice of quadrature rule is dependent on the domain $\Omega$ and $\mu$ (which is in turn application dependent) and should consist of quadrature points that allow for stable integration.

For non-periodic kernels (which we use) this typically implies quadrature points that are clustered towards the boundary $\partial\Omega$. To accomplish this, we *tesselated* $\Omega$ with a simplicial mesh that divided $\Omega$ into some set of nonoverlapping subdomains $\Omega_\ell, \ell = 1, \ldots, N_\Omega$ such that

$$\int_\Omega K(x,y)f(y)d\mu(y) = \sum_{\ell=1}^{N_\Omega} \int_{\Omega_\ell} K(x,y)f(y)d\mu(y). \tag{9}$$

Following standard scientific computing practices Karniadakis & Sherwin (2005); Cantwell et al. (2015) we discretized (9) using a quadrature rule for each of the subdomains $\Omega_\ell$ affinely-mapped from a symmetric quadrature rule on a standard ("reference") simplex $\Omega_{ref}$ in $\mathbb{R}^d$ Freno et al. (2020); see Figure 2. This simplified to the Gauss-Legendre rule in 1D. In Section 3.2.3, we also present results on a 3D problem within the unit ball that utilized a quadrature rule specially tailored for that domain. We further discuss the computational complexity of quadrature in Appendix A.2.

### 2.4.1 CROSS-CHANNEL AFFINE TRANSFORMATIONS

As in other neural operators Li et al. (2021), we also augmented each layer of the KNO with a cross-channel affine transformation (*i.e.*, an MLP dense layer), sometimes called a "pointwise convolution". The output of this operation is added to the output of the integral operator. Formally, we use the modified integral operators that explicitly act on and output vectors of function evaluations on $X_Q := \{y_i^q\}_{i \in [N_Q]}$:

$$\left(\tilde{\mathcal{I}}_q^p\right)_\ell \tilde{\boldsymbol{g}}_\ell = \tilde{\boldsymbol{g}}_\ell W_\ell + \mathbf{1}_{N_Q}(b_\ell) + \left(\left(\mathcal{I}_q^p\right)_\ell \tilde{\boldsymbol{g}}_\ell\right)\big|_{X_Q}, \qquad \ell \in [L-1] \tag{10}$$

$$\left(\tilde{\mathcal{I}}_q^p\right)_L \tilde{\boldsymbol{g}}_L = \left(\left(\mathcal{I}_q^p\right)_L \tilde{\boldsymbol{g}}_L\right)\big|_{X_T}, \tag{11}$$

where $\tilde{\boldsymbol{g}}_\ell \in \mathbb{R}^{N_Q \times p}$ denotes evaluations of the function $\boldsymbol{g}_\ell : \Omega \to \mathbb{R}^p$ on $X_Q$, and $W_\ell \in \mathbb{R}^{p \times p}$ and $b_\ell \in \mathbb{R}^{1 \times p}$ are trainable weights. Note that we abuse notation in the term $\left(\left(\mathcal{I}_q^p\right)_\ell \tilde{\boldsymbol{g}}_\ell\right)\big|_{X_Q}$ by passing the vector $\tilde{\boldsymbol{g}}$ evaluated at quadrature points to the integral operator (rather than a function). The final discretized integral operator outputs values on the training grid $X_T$ for use in evaluating the loss. We found that removing these pointwise convolutions entirely was detrimental to accuracy.

### 2.4.2 LIFTING AND PROJECTION OPERATORS

As with other neural operators, we used standard multilayer perceptrons (MLPs) to parameterize the lifting and projection operators $\mathcal{L}$ and $\mathcal{P}$ that act on discretized inputs. Our lifting operator $\mathcal{L}$ is given by $\mathcal{L}f = \sigma\left(\left(f|_{X_Q} \oplus X_Q\right)W + \mathbf{1}_{N_Q}b\right)$, where $\oplus$ indicates concatenation, $W \in \mathbb{R}^{(d_u + \dim(\Omega_u)) \times p}$ and $b \in \mathbb{R}^{1 \times p}$ are trainable, $\sigma$ is an activation function, and $X_Q$ now represents a matrix of quadrature points. An MLP was also used to parameterize the projection operator $\mathcal{P}$ that combines all the $p$ channels of the hidden layers to produce a single approximation of the output function(s). This MLP consisted of two consecutive $p$-width dense layers ($\mathcal{A} : \mathbb{R}^p \to \mathbb{R}^p$) with nonlinear activation functions and one dense layer with width equal to $d_y$ ($\mathcal{A} : \mathbb{R}^p \to \mathbb{R}^{d_y}$) that did not use an activation function. We use the GeLU activation function in all cases Hendrycks & Gimpel (2023); see Appendix A.8 for more details. In summary, the discretized KNO $\widetilde{\mathcal{H}}$ that we used to numerically construct $\mathcal{H}$ in (2) can be written as a function that takes in $f_{X_T}$ and returns an approximation to the output function $\mathcal{H}(f)$ evaluated at $X_T$:

$$\widetilde{\mathcal{H}}(f_{X_T}) = \left(\mathcal{P} \circ \sigma \circ \left(\tilde{\mathcal{I}}_q^p\right)_L \circ \sigma \circ \left(\tilde{\mathcal{I}}_q^p\right)_{L-1} \circ \sigma \circ \ldots \sigma \circ \left(\tilde{\mathcal{I}}_q^p\right)_1 \circ \mathcal{L}\right)(f_{X_T}) \tag{12}$$

## 3 RESULTS

We now describe our numerical experiments with KNOs and other state-of-the-art neural operators on benchmark problems obtained from Lu et al. (2022). We present results on both tensor-product domains (all of which used boundary-anchored equidistant grids) and irregular domains (which used triangle meshes or point clouds). The KNO models were all trained using the Adam optimizer Kingma & Ba (2017) with a cyclic cosine annealing learning rate schedule. Other technical details are described in Appendices A.6–A.8; we also defer the description of the Advection (I) problem to Appendix A.1. We measured the accuracy of our KNOs by computing the mean and standard deviation of the $\ell_2$ relative errors of each KNO obtained from nine different training runs: three

Table 1: Percent $\ell_2$ relative errors. All non-KNO errors were reported from the literature Lu et al. (2022); Batlle et al. (2024). The last two rows correspond to irregular domains; these used the dgFNO+ rather than the FNO.

| PDE | KM | DeepONet | POD-DeepONet | FNO | KNO |
|---|---|---|---|---|---|
| Burgers' Equation | 2.15 | $2.15 \pm 0.09$ | $1.94 \pm 0.07$ | $1.93 \pm 0.04$ | $\mathbf{0.52 \pm 0.08}$ |
| Advection (I) | $\mathbf{2.15e{-}13}$ | $0.22 \pm 0.03$ | $0.04 \pm 0.00$ | $0.66 \pm 0.10$ | $0.015 \pm 0.01$ |
| Navier-Stokes | – | $1.78 \pm 0.02$ | $1.71 \pm 0.03$ | $1.81 \pm 0.02$ | $\mathbf{1.02 \pm 0.15}$ |
| Darcy (Continuous) | – | $1.36 \pm 0.12$ | $1.26 \pm 0.07$ | $1.19 \pm 0.05$ | $\mathbf{0.91 \pm 0.05}$ |
| Darcy (PWC) | 2.75 | $2.91 \pm 0.04$ | $2.32 \pm 0.03$ | $2.41 \pm 0.03$ | $\mathbf{1.57 \pm 0.06}$ |
| Darcy (triangular) | – | $0.43 \pm 0.02$ | $0.18 \pm 0.02$ | $1.00 \pm 0.03$ | $\mathbf{0.12 \pm 0.01}$ |
| Darcy (triangular-notch) | – | $2.64 \pm 0.02$ | $1.00 \pm 0.00$ | $7.82 \pm 0.03$ | $\mathbf{0.55 \pm 0.04}$ |
| 3D reaction-diffusion | – | $0.127 \pm 0.03$ | $9.40 \pm 8$ | $\mathbf{0.047 \pm 0.02}$ | $0.059 \pm 0.01$ |

separate train/test splits, each with three different random model parameter initializations. These errors were compared to those of DeepONets, POD-DeepONets, and FNOs all as reported in Lu et al. (2022), and kernel/GP-based methods (denoted KM) as reported in Batlle et al. (2024). For the 3D reaction-diffusion problem, we tested DeepONet, POD-DeepONet, and FNO in-house, averaging over five random seeds. See Appendix A.9 for training and architectural details. All errors are reported in Table 1, and all parameter counts are given in Table 2. We used the normalization procedure described in (Lu et al., 2022, Section 3.4) in all cases except the KM.

### 3.1 TENSOR-PRODUCT DOMAINS

#### 3.1.1 BURGERS' EQUATION

We first considered Burgers' equation in one dimension with periodic boundary conditions:

$$\frac{\partial u}{\partial t} + u \frac{\partial u}{\partial x} = \nu \frac{\partial^2 u}{\partial x^2}, \quad x \in (0,1), \quad t \in (0,1),$$

with the viscosity coefficient fixed to $\nu = 0.1$. Specifically, we learned the mapping from the initial condition $u(x,0) = u_0(x)$ to the solution $u(x,t)$ at $t = 1$, *i.e.*, $\mathcal{G} : u_0 \mapsto u(\cdot, 1)$. The input functions $u_0$ were generated by sampling $u_0 \sim \mu$, where $\mu = \mathcal{N}(0, 625(-\Delta + 25I)^{-2})$ with periodic boundary conditions, and the Laplacian $\Delta$ was numerically approximated on $X_T$. The solution was generated as described in (Li et al., 2021, Appendix A.3.1). The full spatial resolution of this dataset was 8192, but the models were trained and evaluated on input-output function pairs both defined on the same downsampled 128 grid (as were the errors). 1000 examples were used for training and 200 for testing. The KNO showed the best accuracy of all the models (Table 1) and achieved roughly a **four-fold improvement** over the next best model (the FNO), while requiring an order of magnitude fewer parameters than the FNO (Table 2).

#### 3.1.2 THE INCOMPRESSIBLE NAVIER-STOKES EQUATIONS

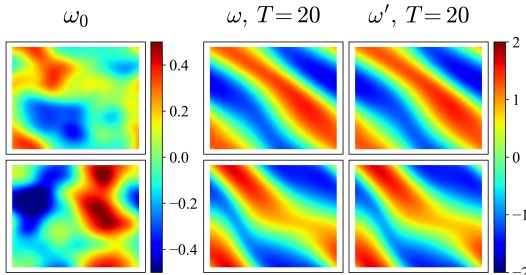

Figure 3: Solutions of the Navier-Stokes problem 3.1.2 on a test example. We show the initial vorticity (left), the solution at $t = 20\Delta t$ (center), and the prediction at $t = 20\Delta t$ (right).

Table 2: Parameter counts for the models in Table 1 provided wherever available. For some we made conservative estimates (detailed in Appendix A.9.2), which are marked with an asterisk. The number of KNO parameters is determined by the hyperparameter choices detailed in Table 4.

| PDE | DeepONet | POD-DeepONet | FNO | KNO |
|---|---|---|---|---|
| Burgers' Equation | 148,865 | 53,664 | 287,425 | 34,307 |
| Advection (I) | – | 86,054 | – | 30,083 |
| Darcy (PWC) | 715,777 | 631,155 | 1,188,353 | 6,723 |
| Darcy (Continuous) | – | – | – | 26,179 |
| Navier-Stokes Equations | – | – | *414,517 | 7,011 |
| Darcy (triangular) | *88,777 | 50,208 | *532,993 | 25,731 |
| Darcy (triangular-notch) | 88,777 | 230,796 | 532,993 | 25,507 |
| 3D reaction-diffusion | 645,120 | 588,928 | 11,952,673 | 26,499 |

In this test, we learned a solution operator for the 2D incompressible Navier-Stokes equations given in vorticity-velocity form on the spacetime domain $[0,1]^2 \times [0,T]$:

$$\frac{\partial \omega}{\partial t} + \mathbf{u} \cdot \nabla \omega = \nu \Delta \omega + f, \ \ \omega(x,0) = \omega_0(x),$$

where $\omega(x,y,t)$ is the fluid vorticity, $\mathbf{u}(x,y,t)$ is the velocity, $\nu = 0.001$ is the viscosity, and $\nabla \cdot \mathbf{u} = 0$; we enforced periodic boundary conditions on $\omega$. The forcing term $f$ was prescribed to be

$$f(x,y) = 0.1 \sin(2\pi(x+y)) + 0.1 \cos(2\pi(x+y)).$$

We learned the mapping from the set of functions $\{\omega(x,y,j\Delta t)\}$, $j = 0,\ldots,9$ to the function $\omega(x,y,20\Delta t)$ by passing the first ten steps as a vector-valued input to the KNO. The input functions were generated by sampling as $\omega_0 \sim \mathcal{N}(0, 7^{3/2}(-\Delta + 49I)^{-2.5})$, and a numerical solution was obtained as in (Li et al., 2021, Section A.3.3). These functions were downsampled from $256^2$ to a resolution of $64^2$ for training and evaluation. We used 1000 examples for training and 200 for testing. Once again, the KNO outperformed all other models (Table 1) while requiring fewer than 10k trainable parameters (Table 2).

### 3.1.3 DARCY FLOW

We used KNOs to learn two operators $\mathcal{G} : K \mapsto h$ associated with 2D Darcy flow

$$-\nabla \cdot (K(x,y)\nabla h(x,y)) = f(x,y), \quad (x,y) \in \Omega.$$

on the $\Omega = [0,1]^2$. For case (1), the permeability field was generated via $K = \psi(\mu)$, where $\mu \sim \mathcal{N}(0, (-\Delta + 9I)^{-2})$, and $\psi$ is a function that pointwise converts all non-negative values to 12 and all negative values to 3. We henceforth refer to this problem as "Darcy (PWC)". Case (2) involved generating continuous permeability fields using a Gaussian process parameterized with a zero mean and Gaussian covariance kernel; see Li et al. (2021) for details. We refer to this problem as "Darcy (cont.)". Both problems used 1000 training functions and 200 test functions. The Darcy (PWC) training functions were computed on a $421^2$ grid Lu et al. (2022) and subsampled to a $29^2$ grid. The Darcy (cont.) solutions were obtained using the Matlab PDE Toolbox on an unstructured mesh with 1,893 elements, with Neumann and Dirchlet boundary conditions were imposed on the top and bottom boundaries, and the left and right boundaries respectively. The solutions $h$ were then linearly interpolated from the mesh to the same uniform $20^2$ grid upon which $K$ was originally defined so that both functions shared the same discretization.

The KNO achieved under $1\%$ error on the Darcy (cont.) problem, once again showing the best accuracy among all the neural operators tested. Further, in Darcy (PWC), the KNO achieved a $30\%$ lower error than the second-best model (FNO) while requiring *over two orders of magnitude fewer trainable parameters than FNO and DeepONet* and *almost two orders of magnitude fewer trainable parameters than POD-DeepONet*.

### 3.2 IRREGULAR DOMAINS

#### 3.2.1 DARCY (TRIANGULAR)

We also examined two Darcy flow problems where the input and output functions were both discretized on an irregular spatial domain. Specifically, as in Lu et al. (2022), we learned the mapping

from the Dirichlet boundary condition to the pressure field over the entire domain, *i.e.*, the operator $\mathcal{G} : h(x,y)|_{\partial\Omega} \mapsto h(x,y)$. We report the dgFNO+ variant's performance under the FNO column since it can tackle both irregular geometries and different input and output domains. Here $K(x,y) = 0.1$ and $f = -1$. The input functions $h(x,y)|_{\partial\Omega}$ for both problems were generated as follows. First, we generated $\tilde{h}(x) \sim \mathcal{GP}(0, \mathcal{K}(x,x'))$, $\mathcal{K}(x,x') = \exp[-\frac{(x-x')^2}{2l^2}]$, where $l = 0.2$ and $x, x' \in [0,1]$. We then simply evaluated $\tilde{h}(x)$ at the $x$-coordinates of the boundary points of each unstructured mesh to obtain $h(x,y)\big|_{\partial\Omega}$. The Matlab PDE Toolbox was used both to generate unstructured meshes and numerical solutions Lu et al. (2022). Both problems used 1900 training examples and 100 test examples.

This problem utilized an 861 vertex unstructured mesh with 120 points lying on the boundary; see Lu et al. (2022) (Figure S2 (c)). Once again, the KNO showed the best accuracy of all neural operators on this domain, partly illustrating the effectiveness of our quadrature rule (see Section 2.4). As in the other test cases, the KNO required far fewer trainable parameters than existing neural operators.

### 3.2.2 DARCY (TRIANGULAR-NOTCH)

This problem involved removing a small notch from the triangular domain Lu et al. (2022) (see Figure 4). The mesh contained 2,295 vertices with 260 of those on the boundary. Again, the KNO outperformed the other models; it was almost twice as accurate as the next best model, the POD-DeepONet, with an order of magnitude fewer parameters than dgFNO+. The results here underscore KNO's flexibility, both in handling different input and output spaces and in tackling irregular geometries.

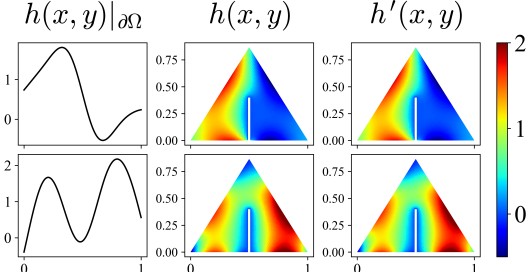

Figure 4: Solutions of the Darcy (triangular-notch) problem 3.2.2. We show two input functions (left), solution functions (middle), and the KNO predictions (right).

### 3.2.3 3D REACTION-VARIABLE-COEFFICIENT-DIFFUSION

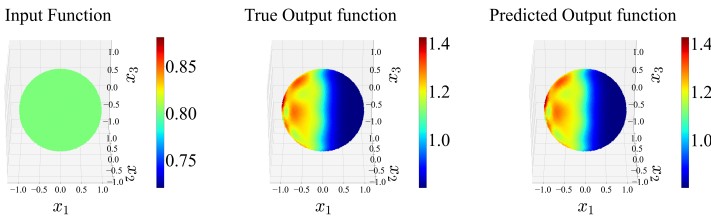

Figure 5: The 3D reaction-diffusion problem 3.2.3, where an input function is given (left), the true output function (center), and a prediction from the KNO (right).

Finally, we investigated a 3D problem reaction-diffusion problem in the unit ball, (*i.e.*, the interior of the unit sphere) where a chemical with concentration $c(y,t)$ is governed by:

$$\frac{\partial c}{\partial t} = k_{\text{on}}\left(R - c\right) c_{\text{amb}} - k_{\text{off}}\, c + \nabla \cdot \left(K(y)\nabla c\right), \; y \in \Omega, \; t \in [0, 0.5],$$

where $y = (y_1, y_2, y_3)$ and $K(y)\frac{\partial c}{\partial n} = 0$ on $\partial\Omega$. Here, $R = 2.0$ throttles the reaction, and the $k_{\text{on}}$ and $k_{\text{off}}$ are discontinuous reaction constants that introduce a sharp solution gradient at $y_1 = 1.0$:

$$k_{\text{on}} = \begin{cases} 2, & y_1 \leq 1.0, \\ 0, & \text{otherwise,} \end{cases} \qquad k_{\text{off}} = \begin{cases} 0.2, & y_1 \leq 1.0, \\ 0, & \text{otherwise.} \end{cases}$$

The diffusion coefficient is also a spatially varying function with a steep gradient given by:

$$K(y) = B + \frac{C}{\tanh(A)}\left((A - 3)\tanh(8x - 5) - (A - 15)\tanh(8x + 5) + A\tanh(A)\right),$$

where $A = 9$, $B = 0.0215$, and $C = 0.005$. $c_{\text{amb}} = (1 + \cos(2\pi y_1)\cos(2\pi y_2)\sin(2\pi y_3))e^{(-\pi t)}$ is a background source of chemical accessible for reaction. We set the initial condition to be $c(y, 0) \sim \mathcal{U}(0, 1)$, and learned the solution operator $\mathcal{G} : c(y, 0) \to c(y, 0.5)$. The PDE was solved on 4325 collocation points using a 4th-order accurate RBF-FD solver Shankar & Fogelson (2018) to generate 1000/200 train and test input/output function pairs, respectively. For the dgFNO+, we used a $16^3$ uniform grid. The KNO attained twice the accuracy of vanilla DeepONet and two orders of magnitude greater accuracy than the POD-DeepONet, and had comparable accuracy to the "dgFNO+" method despite using three orders of magnitude fewer parameters.

## 3.3 RUNTIME COMPARISON

We also present training and inference times for the KNO and FNO in Table 3; for the FNO, we present timings for test problems where the architecture is known. The KNO was implemented in Jax while the reference FNO was implemented in PyTorch. The KNO is generally faster than the FNO, with further potential for speedups through the use of optimized sparse matrix operations.

Table 3: Average training time per epoch and average inference time (both in seconds) on the test set over $20,000$ epochs measured on a NVIDIA GeForce RTX 4080. Datasets are not mini-batched.

| PDE | Training time | | Inference time | |
|---|---|---|---|---|
| | **FNO** | **KNO** | **FNO** | **KNO** |
| Burgers' Equation | 1.40e–2 | 5.04e–3 | 1.38e–3 | 6.51e–4 |
| Advection (I) | – | 2.16e–3 | – | 7.27e–4 |
| Darcy (PWC) | 8.72e–2 | 4.85e–2 | 4.39e–3 | 4.36e–3 |
| Darcy (Continuous) | – | 4.08e–2 | – | 2.68e–3 |
| Navier-Stokes Equations | *2.85e–1 | 1.42e–1 | *2.89e–2 | 2.17e–2 |
| Darcy (triangular) | *4.30e–1 | 8.52e–2 | *5.00e–3 | 1.44e–3 |
| Darcy (triangular-notch) | 4.30e–1 | 2.00e–1 | 5.00e–3 | 4.85e–3 |
| 3D reaction-diffusion | 6.54e–1 | 4.79e–1 | 3.66e-2 | 4.33e–2 |

## 4 CONCLUSION

We presented the kernel neural operator (KNO), a novel, simple, and transparent architecture that leverages kernel-based deep integral operators discretized by numerical quadrature. The use of explicit, closed-form, diagonal, matrix-valued kernels allowed the KNO to achieve superior accuracy with far fewer trainable parameters than other neural operators (on both regular and irregular domains). We found that compactly-supported kernels used throughout (save the final layer) were the optimal choice to obtain a general purpose architecture well-suited to a wide variety of operator learning problems. In our view, our results also indicate that it may be possible to achieve similar parameter counts (and possibly relative errors) with other neural operators such as DeepONet and the FNO, albeit with architecture tuning, careful training, and problem-specific initializations.

For future work, we will prove the universal approximation capabilities of the KNO and leverage the closed form kernels to derive rigorous error estimates for the approximation of PDE solution operators. We will also explore interpretable lifting and projection operators, problem-specific architectures (for instance, for linear operators), novel quadrature schemes, and other types of problem-dependent kernels not discussed in this work. We anticipate that the KNO will be widely applicable to a variety of machine learning tasks beyond approximating PDE solution operators. We plan to explore these in future work as well.

**Ethics Statement**: To the best of the authors' knowledge, there are no negative societal impacts of our work including potential malicious or unintended uses, environmental impact, security, or privacy concerns.

**Reproducibility Statement**: We include all source code, datasets, and run files and instructions to faithfully reproduce the results of our experiments in the supplementary portion of our submission. We also include training and architectural details of KNO and other models in appendices A.8.3 and A.9, respectively.

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

# A  APPENDIX

## A.1  THE ADVECTION EQUATION

Our results in the main body of the paper also include an operator learning problem associated with the 1D advection equation, given by

$$\frac{\partial u}{\partial t} + \frac{\partial u}{\partial x} = 0, \quad x \in [0, 1], \quad t \in [0, 1],$$

with a periodic boundary condition $u(0, t) = u(1, t)$. We learned the mapping $\mathcal{G} : u_0 \mapsto u(\cdot, 0.5)$ (Lu et al., 2022, Case (I), Section 5.4.1). The initial condition was a square wave with center, width, and height uniformly sampled from $[0.3, 0.7]$, $[0.3, 0.6]$, and $[1, 2]$ respectively. The spatial resolution for this data was fixed to 40, and we generated 1000 training and testing examples. The KNO again outperformed all the neural operators (Table 1), but was unable to match the kernel method (KM), which used a linear kernel to recover the linear operator $\mathcal{G}$. We believe it should be possible to obtain the same accuracy with the KNO by removing nonlinearities as appropriate; however, we leave an exploration of problem-specific architectures for future work and focus on the generalizable and flexible architecture reported here.

## A.2  COMPLEXITY OF COMPUTING INTEGRALS VIA QUADRATURE

We now discuss the complexity of evaluating the kernel integrals using quadrature, in contrast with The FNO's approach which leverages the FFT and admits $O(n \log n)$ per integral (following the runtime comparisons reported in section 3.3). If $n$ is the size of the input sample grid, the cost of quadrature is dependent on the number of quadrature points. Our approach, that is, to precompute a quadrature rule with $N_q$ points and weights incites a cost of $O(N_q)$ per output location; thus, incites a total cost of $O(nN_q)$. Now in practice, $N_q < n$ and also with the KNO, $n = N_q$ for $\ell \in [L - 1]$ so that the total cost is closer $O(N_q)$ for a given $n$, which makes this competitive with the $O(n \log n)$ FNO cost. Regardless, both the FNO and the KNO likely suffer from the curse of dimensionality on their respective grids and their costs are dominated by MLP operations. The quadrature and polynomial approximation literature contains many approaches to tackle this (sparse grids, composite quadrature rules, and so forth). We plan to tackle this in future work.

However, it is important to keep sight of the fact that quadrature allows us to tackle problems with irregular domain geometries and point cloud data, unlike the standard FFT used in the FNO; the alterative would be to use deformation maps to learn coordinate transforms to regular grids Li et al. (2023; 2024), but such mappings do not always exist. Further, it may be possible to accelerate our framework by mapping KNO layers to standard convolutional layers. This could be done by imposing structure on the feature detector and/or filter to mimic the operation of the kernel in order to further leverage existing ML toolchains (while losing geometric flexibility as in the FNO). Similar work was done on the function approximation side in integrated neural networks (INNs) Solodskikh et al. (2023). It may also be of interest to explore things in the other direction: engineering kernels and/or loss functions that mimic the effects of certain convolutional layers (say, specific kinds of filters, stencils/filter sizes, and feature detectors).

## A.3  VARYING THE MATRIX-VALUED KERNEL'S STRUCTURE

Our decision to parameterize the KNO's integral operators with a diagonal matrix-valued kernel was made with the intention of developing a parameter-efficient neural operator that performed on-par with or better than existing neural operator on the various benchmark datasets. Nonetheless, the KNO allows us to adopt other formulations of this matrix-valued kernels, and it is unclear if this is possible in FNOs. This is particularly relevant in the context of PDEs, where the solution operators can be expressed in terms of Green's functions that themselves have inherent structure Boullé & Townsend (2023).

For completeness, we present some preliminary experiments with a KNO whose integral operators were parameterized by a diagonal matrix-valued kernel, a tridiagonal matrix-valued kernel, and a fully dense matrix-valued kernel, respectively, on Burgers' equation and the Darcy (cont) problem 6 given a fixed training configuration i.e. $p = q = 32$, $L - 1 = 3$, 30, 000 epochs, and the same number of quadrature nodes as reported in 4. The results show that the optimal structure of the matrix-valued kernel may be application dependent; note that the tridiagonal one performed better on the Darcy (cont) problem and the dense one performed best on the Burgers' equation. A cautious reader might be skeptical as to why a diagonal-matrix valued kernel was effective, given it does not couple

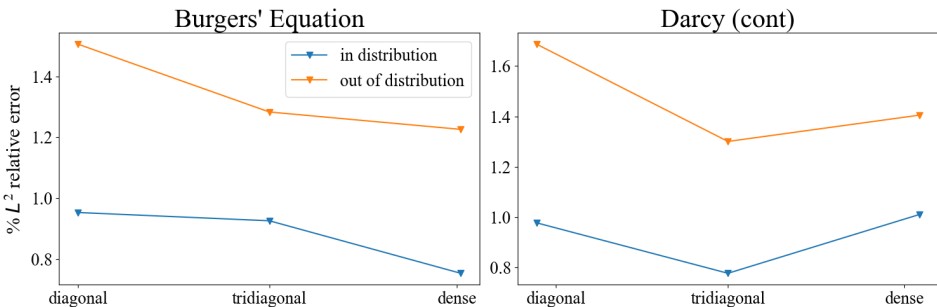

Figure 6: KNO results for two problems with its integral operators parameterized by three different matrix-valued kernel variants.

information across channels. The answer lies in the pointwise convolutions present in both FNOs and KNOs, which serves this purpose. In KNOs, we found that the architectural choice of one kernel per channel (or fewer) combined with pointwise convolution coupling across channels resulted in high accuracy with a lower parameter count (and a simple architecture). Such a choice in the FNO is likely to reduce parameter counts but also reduce accuracy, since the FNO implicitly imposes both periodicity and denseness in its matrix-valued kernels. We hypothesize that the KNO kernels are learning information "local" to channels, and pointwise convolutions then couple information across channels in a more global fashion. We plan to explore these details in a follow-up paper.

### A.4 Zero-shot super-resolution

As every layer in the KNO is composed of function-space operations, the KNO can achieve zero-shot super resolution, *i.e.,* it can produce operator solutions at arbitrary resolutions without retraining, much like the FNO. This is visualized in Figure 7.

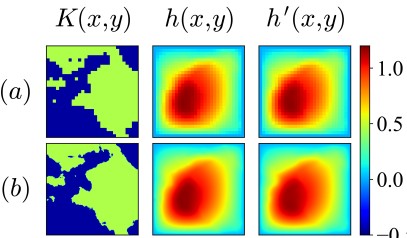

Figure 7: An illustration of zero-shot super-resolution. The KNO was trained on the Darcy (PWC) dataset using a $29 \times 29$ grid (row a). It was then evaluated at a resolution of $211 \times 211$ (row b). We show the permeability field input (left), the actual pressure field (middle), and the predicted pressure (right).

### A.5 Other kernel choices

As mentioned previously, we also explored the use of other kernels, enumerated below, within our integral operators, however the KNO architecture reported in the main text out-performed all of the other kernels tested.

1. Gaussians everywhere (overfitting): When isotropic Gaussian kernels $\phi(x, x') = e^{\epsilon^2 \|x-y\|_2^2}$ were used throughout the KNO, we found that the resulting architecture tended to achieve low training error and high test error, while also being highly sensitive to the initial random seed used to optimize the KNO.

2. Wendland everywhere (higher training and test errors): When we used Wendland kernels everywhere, we found that the resulting architecture had significantly higher training and test errors than using Wendland kernels almost everywhere and a spectral mixture kernel at the end. This experiment revealed to us that using a kernel that was not compactly-supported for the final integral operator was important for accuracy. This is possibly due to

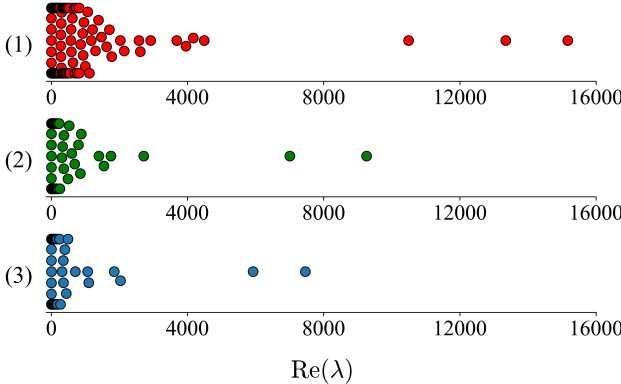

Figure 8: Eigenvalues of the neural tangent kernel (NTK) for three choices of kernels: (1) Gaussian kernels for $\left(\mathcal{I}_q^p\right)_k$, $k = 1, \ldots, L$; (2) $C^4(\mathbb{R}^3)$ Wendland kernels for $\left(\mathcal{I}_q^p\right)_k$, $k = 1, \ldots, L-1$ and a Gaussian kernel for $\left(\mathcal{I}_q^p\right)_L$; and (3) $C^4(\mathbb{R}^3)$ Wendland kernels for $\left(\mathcal{I}_q^p\right)_k$, $k = 1, \ldots, L-1$ and a Gaussian spectral mixture kernel for $\left(\mathcal{I}_q^p\right)_L$.

> the fact that our final integral operator simply did not use a cross-channel affine transformation (aka pointwise convolution).

3. Wendland almost-everywhere, Gaussian for $\left(\mathcal{I}_q^p\right)_L$: This choice of kernels produced excellent training and test accuracy and was relatively robust to choices in the other hyperparameters, but produced higher errors than using the spectral mixture kernel for $\left(\mathcal{I}_q^p\right)_L$.

In order to quantify the differences between these choices, we computed the eigenvalue spectra of the neural tangent kernel (NTK) matrix for the final KNO architecture, for cases (1) and (3) above; case (2) produced reasonable spectra but lowered accuracy (not shown). The spectra of these NTK matrices are shown in Figure 8; in general, more tightly clustered eigenvalues of the NTK matrix are indicative of fewer local minima and a lower tendency to overfit. We see that the Gaussian results in a spectrum with a very large range, while the Wendland + Gaussian choice results in a much tighter spectrum; the Wendland + spectral mixture choice results in the tightest spectrum of all. It is possible that stable kernel evaluation via a Hilbert-Schmidt decomposition might improve the Gaussian's NTK spectra Fasshauer & McCourt (2015), but we save such an exploration for future work.

We also believe Wendland kernels were vital in the kernel interpolant that transfers data to the quadrature points as their finite smoothness and corresponding sparse interpolation matrices allowed us to avoid the exponential ill-conditioning inherent to interpolation on boundary-anchored equispaced grids. The Gaussian kernel, on the other hand, is infinitely-smooth and capable of exponential convergence on infinitely-smooth target functions. Its corresponding linear system hence suffers from exponential ill-conditioning (much like polynomial Vandermonde matrices); this follows directly from the impossibility theorem Platte et al. (2011); Adcock et al. (2019).

We also ran another experiment (results not shown) to investigate the impact of limited smoothness of the Wendland kernels on efficacy. Specifically, we replaced the Wendland kernels with $C^4(\mathbb{R}^3)$ Matérn kernels, which are finitely-smooth but *not* compactly-supported. We observed worse errors in all our experiments using Matérn kernels over Wendland kernels (but still better results than using the Gaussian everywhere). It may be possible to understand this in terms of the Fourier transforms of these kernels. In general, in the context of interpolation, the rate of decay of the Fourier transform of a kernel can affect its approximation power Fasshauer (2007). In this context, we believe it affects trainability also. Wendland kernels, being compactly-supported, have Fourier transforms with heavy frequency tails (by the Fourier uncertainty principle), thus carrying more information. In contrast, Gaussians and even other less smooth Matérn kernels have more concentrated Fourier transforms with fast decay (exponential in the frequency for Gaussian kernels, algebraic for the Matérn kernels), which likely results in a loss of information during training. In future work, we plan to apply Fourier analysis tools to further understand and clarify this intuition.

### A.6 KNO SPARSITY

We also tracked the learned sparsity in the KNOs, specifically the average number of zeros in each kernel evaluation matrix formed by the Wendland kernels. This metric roughly converged to $20\%$, $20\%$, $26\%$, $27\%$ and $23\%$ for the tensor-product domain datasets in the order by which they are listed in Table 1. Interestingly, for the Darcy problems on irregular domains, we observed lower sparsity percentages, $6\%$ and $4\%$, for the triangular and triangular-notch problems respectively. It is possible that this was because the triangular Darcy problems involved mapping boundary conditions to solutions over the full domain. Lastly, sparsity on the 3D diffusion-reaction problem converged to approximately $60\%$. As the solution functions exhibited sharp solution gradients in the center of the sphere, we speculate that this is due to kernels focusing on this area, where such sharp gradients need to be more accurately resolved, but we leave a deeper exploration of the connection between sparsity and the operator learning problem for future work.

### A.7 ABLATION STUDIES

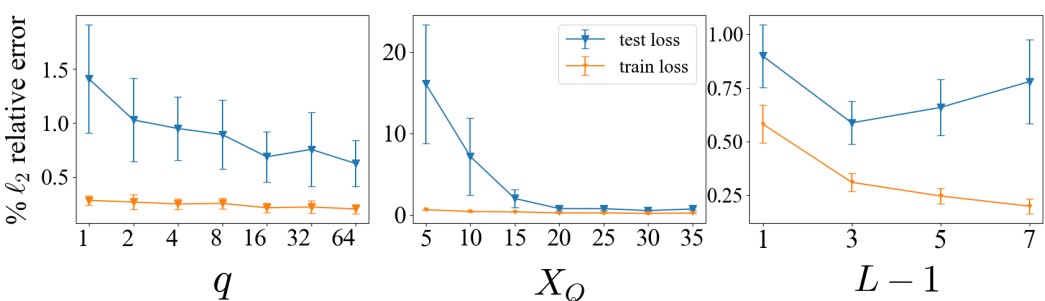

Figure 9: Ablation Study for Burgers' Equation. On the left the number of trainable kernels $q$ per integration block (for $p = 64$) was systematically varied with a constant architecture otherwise ($X_Q = 30$ and $L - 1 = 6$). The number of Gauss-Legendre quadrature points (center) were scaled in the same capacity with $p$ and $q$ fixed to $64$ and $L - 1 = 6$. The depth (right) was also scaled with $p, q = 64$ and $X_Q = 30$.

To verify the robustness of our results under training, we also conducted ablation studies on Burgers' equation. We focused on the ratio between the number of trainable kernels $q$ as compared and the channel lift size $p$, on the number of Gauss-Legendre quadrature points employed, and on the model depth; that is, the total number of integration blocks excluding the evaluation block ($L - 1$). The results are shown in Figure 9.

Figure 9 (left) shows that while the best results are obtained with $q = p$, smaller values of $q$ may also suffice, *i.e.*, one may be able to use fewer trainable parameters than channels, allowing for significant reductions in computational cost. It is also likely that this can be done with the FNO family of neural operators. Figure 9 (middle) also shows a relative insensitivity of our results to the number of quadrature points for the datasets used in this work; however, it is not unreasonable to expect some relationship between the number of spatial samples of the input and output functions and the number of quadrature points. We plan to explore this connection in future work. Finally, Figure 9 (right) shows that the depth of the KNO was much more important, especially for generalization. KNOs with more layers tended to overfit on this 1D problem. However, it is plausible that there is an optimal depth for a given dataset in a particular spatial dimension. We leave such an exploration for future work also.

### A.8 IMPORTANT ARCHITECTURAL AND TRAINING DETAILS FOR THE KNO

#### A.8.1 INITIALIZATION AND REGULARIZATION

We initialized all trainable parameters associated with kernels by sampling $\mathcal{N}(1, 0.01)$ and applied a softplus transform to enforce that all kernel shape parameters were positive. We also include a very mild $\ell_2$ regularization to the shape parameters in the loss term to encourage sparsity but did not find this to substantially impact convergence.

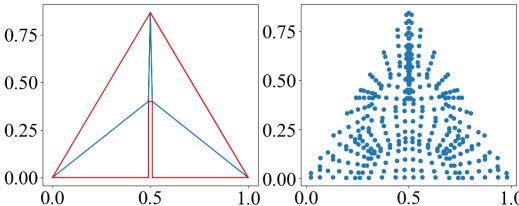

Figure 10: On the right is a quadrature rule for the Darcy (triangular-notch) problem, created by mapping the reference triangle's rule 2 defined at $\left[(0,0),(1,0),(\frac{\sqrt{3}}{2},0.5)\right]$, to a five triangle Delaunay mesh (left) over the domain. The cut out 'notch' is defined by the vertices $[(0.49,0),(0.51,0),(0.49,0.4),(0.51,0.4)]$.

Spherical Volume Quadrature Rule

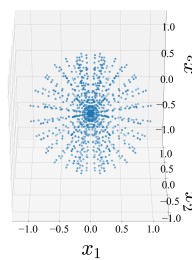

Figure 11: The quadrature rule used for for the 3D reaction-diffusion problem.

### A.8.2 QUADRATURE POINTS

We now briefly present details on the quadrature points used in the different operator learning problems. For the 2D examples, we took the approach of subdividing the domain into some number of triangles, then mapped the integrals on each triangle back to our reference triangle (as was mentioned previously).

1. Also mentioned previously, all 1D examples used Gauss-Legendre points defined on $[-1,1]$. We simply transformed the Gauss-Legendre points to the domain of interest in this case.

2. For the Darcy (PWC) and Navier-Stokes problems, we subdivided the domain $[0,1]^2$ into four squares, then further subdivided each square into two triangles, for a total of eight triangles.

3. For the Darcy (cont.) problem, we simply used two triangles.

4. For the Darcy (triangular-notch) problem, we created a five triangle Delaunay mesh over the whole domain; see Figure 10.

5. For the Darcy (triangle) problem, the domain matched our reference triangle, and so no further subdivision or mapping was used.

### A.8.3 HYPERPARAMETER CHOICES

The optimal hyperparameters for the KNO on each dataset are shown in Table 4. These hyperparameters were tuned manually via trial and error. The following are some relevant observations:

(1) Setting the depth $L - 1 = 4$ was the most reliable choice with a few exceptions, namely the Advection (1) and Burgers' equation problems, where the optimal depth increased to 5 and 6 respectively. Usually, increasing the depth resulted in training instability and/or overfitting. However, it is possible that more complicated residual connections or an addition of batch normalization between integration layers could allow for deeper models to be more successful. The KNO is well-suited to such augmentations since it innately possesses a very small number of trainable parameters per layer.

Table 4: This table denotes our chosen configuration for KNO on each dataset. An asterisk indicates a hyperparameter that when increased also increases the total number of trainable parameters. Here $X_Q$ is the total number of quadrature nodes, $L-1$ is the 'depth' as referred to previously and $q$ is effectively the number of trainable kernels relative to the channel lift dimension $p$.

|  | $X_Q$ | $(L-1)^*$ | $q^*$ | $p^*$ |
|---|---|---|---|---|
| Burgers' Equation | 30 | 6 | 64 | 64 |
| Advection (I) | 32 | 5 | 64 | 64 |
| Darcy (PWC) | 864 | 4 | 16 | 32 |
| Darcy (Continuous) | 294 | 4 | 64 | 64 |
| Navier-Stokes | 384 | 4 | 16 | 32 |
| Darcy (triangular) | 300 | 4 | 32 | 64 |
| Darcy (triangular-notch) | 375 | 4 | 16 | 64 |
| 3D reaction-diffusion | 1000 | 4 | 64 | 64 |

(2) We found that altering the MLP layer width to a value other than $p$ provided no benefit.

(3) In several instances, we were able to reduce $q < p$, which not only reduced trainable parameters, but also provided regularization, slightly improving test accuracy. These problems were: Darcy (PWC) where $q = 16$ and $p = 32$, the Navier-Stokes equations ($q = 16$ and $p = 32$), Darcy (triangular) ($q = 32$ and $p = 64$), and Darcy (triangular-notch) ($q = 16$ and $p = 64$).

(4) On the 1D problems, we observed optimal performance with $\sim 30$ quadrature nodes. In contrast, this number was $\sim 300 - 400$ for the 2D datasets, reflecting the exponential relationship between the number of quadratures nodes and the spatial dimension. A slight exception to this is the Darcy (PWC) problem, in which KNO performed optimally with $\sim 900$ nodes. This is potentially a result of the piecewise constant nature of the input function, which necessitates more quadrature nodes to resolve the discontinuities. Here (and in general) an adaptive, problem specific quadrature rule could be beneficial and potentially enable us to reduce $X_Q$ further. We leave such an exploration for future work.

### A.8.4 TRAINING DETAILS

All models were trained on either an NVIDIA GeForce RTX 2080 Ti or an NVIDIA GeForce RTX 4080. We found that freeze-training (i.e. training kernel-based layers independently back to front) prior to training the full model hastened its convergence and so used this tactic quite often for the sake of convenience. More specifically, for a certain number of epochs, we allowed only a single layer to affect gradient updates, effectively freezing all other layers. We then repeated this process for each layer. Finally, we trained the model while allowing all pretrained layers to contribute to updates. It is highly likely that such training would be beneficial for the FNO family of neural operators also. In fact, a version of this training procedure has already proven effective for DeepONets Peyvan et al. (2024). In Table 5, we report the number of training epochs for each PDE example. The second

Table 5: Number of epochs used in KNO training for different PDE examples.

| PDE | Number of epochs | Number of epochs per layer |
|---|---|---|
| Burgers' Equation | 30,000 | 625 |
| Advection (I) | 70,000 | 2857 |
| Darcy (PWC) | 15,000 | 166 |
| Darcy (Continuous) | 30,000 | 666 |
| Navier-Stokes | 20,000 | 0 |
| Darcy (triangular) | 20,000 | 166 |
| Darcy (triangular-notch) | 5,000 | 83 |
| 3D diffusion-reaction | 30,000 | 0 |

column indicates the number of epochs allocated to each layer during freeze training.

### A.9 Details on other models

Here, we provide or cite architecture details for other models, as recorded in Lu et al. (2022) and the accompanying code. Note that in most cases, we did not implement these models; we merely reported results from Lu et al. (2022) for the neural operators and Batlle et al. (2024) for the kernel method. In the case of the 3D reaction-diffusion problem, however, we did implement and train the models ourselves. Specifically we trained the DeepONet models for 150,000 epochs, calculated over five random seeds, and annealed the learning rates with an inverse-time decay schedule. For the dgFNO+, we followed the training outlined in Li et al. (2021), but doubled the number of epochs.

#### A.9.1 Architectures

**DeepONets**    We reported results for both standard DeepONets and POD-DeepONets in Table 1 directly using the results reported in Lu et al. (2022). The architectural details of those operators are given in (Lu et al., 2022, Section S2, Tables S2 and S3). However, those tables do not report the CNN parameters or architectures for all of their models; we estimated those whenever possible from the accompanying code in `https://github.com/lu-group/deeponet-fno` for parameter counts. For the 3D reaction-diffusion problem, the DeepOnet architecture had 3 layers and 128 nodes in both the branch and trunk net, with $p = 100$, while the POD-DeepONet had the same size branch net, but with $p = 20$ POD bases.

Table 6: FNO/dgFNO+ architecture details.

| PDE | Channel dimension $p$ | Number of Fourier modes retained |
|:---:|:---:|:---:|
| Burgers' | 64 | 16 |
| Darcy (PWC) | 32 | 12 |
| Darcy (triangular notch) | 32 | 8 |
| 3D reaction-diffusion | 32 | 9 |

**FNOs**    Again, we reported results for the FNO and the "dgFNO+" in Table 1 directly using the numbers from Lu et al. (2022). However, that work unfortunately does not describe the FNO or "dgFNO+" architecture in detail. Of the examples used in this paper, the FNO or dgFNO+ code for the Burgers' problem, the Darcy (PWC) case, and the Darcy (triangular-notch) case was available in `https://github.com/lu-group/deeponet-fno/tree/main/src` (under the appropriate subfolder). The code did allow for easy extraction of the channel dimension $p$ and the number of Fourier modes retained after truncation. We report these in Table 6 wherever available.

**Kernel method (KM)**    Finally, we also reported results for the KM in Table 1. These were directly obtained from (Batlle et al., 2024, Table 3) wherever possible: for the Burgers' equation, the Advection (I) problem, and the Darcy (PWC) problem. While Batlle et al. (2024) also contains results for a Navier-Stokes problem, that one was different from ours and so we do not report it here. We also only selected the highest accuracy results from that work, which corresponded to the following kernels on the following problems: the Matérn or rational quadratic (RQ) kernel for the Burgers' equation (both apparently produced similar results); the same kernels for the Darcy (PWC) problem; and finally the linear kernel for the Advection (I) problem (which involved learning a linear operator).

#### A.9.2 Parameter estimates (Table 2)

We took our estimate of the parameter count of the FNO on the Navier-Stokes Equations from the FNO-2D model listed in Table 1 of Li et al. (2021). We believed this was reasonable as that problem was a small variation on the one tested herein. Our estimate for the parameter count of the FNO used in the Darcy (triangular) problem, a dgFNO+ variant, was taken by assuming the same model configuration as in the Darcy (triangular-notch) problem; the latter was reported in Lu et al. (2022). We estimated the DeepONet parameter count on the same problem by assuming the model size and output dimension to be equivalent to the Darcy (triangular-notch) problem (Lu et al., 2022, Table S2)). The KM had the smallest number of trainable parameters: 0 for the linear kernel, and 2 for the Matérn and RQ kernels. These were tuned by cross-validation or log marginal likelihood maximization over the training data (Batlle et al., 2024, Section 4.1.1). Note however that the KM required solving large dense linear systems.

