# OpenReview forum: "Kernel Neural Operators (KNOs) for Scalable, Memory-efficient, Geometrically-flexible Operator Learning"
_ICLR.cc/2025/Conference — ICLR 2025 Conference Withdrawn Submission_

### Official Review · Reviewer_H5JD · 2024-10-28

**Soundness:** 2
**Presentation:** 2
**Contribution:** 2
**Rating:** 5
**Confidence:** 5

**Summary:**

This paper presents Kernel Neural Operators (KNOs), a new method for operator learning that employs deep, kernel-based integral operators with quadrature to approximate function spaces. By utilizing parameterized, closed-form kernels that are finitely smooth and compactly supported, KNOs can reduce the number of trainable parameters compared to neural operators like the Fourier Neural Operator (FNO). Additionally, KNOs are well-suited for handling irregular domains and non-uniform data.

**Strengths:**

- Developed a novel neural operator modeled as a kernel integral operator.
- Proposed a new method of parameterizing kernel integral operators with radial Wendland kernels.
- This approach minimizes the number of trainable parameters in KNO, making it effective with fewer parameters.
- Capable of processing inputs on arbitrary grids.

**Weaknesses:**

- The authors conducted multiple experiments, yet the complexity of these experiments may be limited. For example, the solution operators for 1D PDEs are relatively straightforward, and in the Navier-Stokes experiment, the low Reynolds number leads to a very smooth vorticity field. Given that most of KNO's parameters come from the MLPs within the integral layers, it remains uncertain how much the kernel integral layers contribute or if simple MLPs could approximate the solution operators just as effectively.
- I believe the comparison between KNO and the baselines was not entirely fair. Rather than running their own baseline experiments, the authors relied on results reported in other papers. For many cases, the specific architectures used are unclear, which raises concerns about the transparency and applicability of these comparisons. This issue is especially evident in the FNO model used for the Darcy (triangular-notch) experiment, where only 8 Fourier modes were applied, suggesting suboptimal model selection. Additionally, the FNO model has demonstrated an overall relative L2 error of around 0.8% for an autoregressive task that maps the first 10 time steps to the last 40, step-by-step, using the updated implementation released over two years ago. However, the authors reported a single-step error of 1.81% mapping the first 10 steps to the 20th step for FNO; this is possible, but no details and their own implementations are provided, which raises doubts about the fairness of these comparisons, as essential details to verify the results are missing. For greater transparency, the authors should have conducted their own baseline training and model selection. The authors should include experimental details in greater extend.
- No convolution-based baselines were run, despite their demonstrated effectiveness in learning PDEs (such as U-Net, [1], [2]).
- The experiments conducted do not substantiate the claim that KNO performs well on irregular domains. The authors should compare KNO with additional baselines designed for irregular and more complex domains to better demonstrate its effectiveness (such as [3]).
- The novelty and contribution of this work may be somewhat limited, as the overall framework largely mirrors that of FNO, with the primary differences lying in how the kernel and kernel integral are managed. However, the authors do not provide enough discussions and evidence, either theoretical or empirical to convey their claims.

**Overall this paper presents an interesting idea. If the concerns and questions regarding the experiments can be answered satisfiedly, I would be happy to increase my score.**

[1] Towards Multi-spatiotemporal-scale Generalized PDE Modeling; Jayesh K. Gupta, Johannes Brandstetter; 2022

[2] Convolutional Neural Operators for robust and accurate learning of PDEs; Bogdan Raonić, Roberto Molinaro, Tim De Ryck, Tobias Rohner, Francesca Bartolucci, Rima Alaifari, Siddhartha Mishra, Emmanuel de Bézenac; 2023

[3] Operator Learning with Neural Fields: Tackling PDEs on General Geometries; Louis Serrano, Lise Le Boudec, Armand Kassaï Koupaï, Thomas X Wang, Yuan Yin, Jean-Noël Vittaut, Patrick Gallinari; 2024

**Questions:**

- Why didn’t the authors conduct their own experiments? It is important given that the reported results, e.g. for NS, are from a paper published in 2021 (on Arxiv 2021), updates to the FNO architecture since then have led to notable improvements.
- Can the proposed architecture be implemented via standard convolutional layers if the input domain is sampled on a regular grid?
- The experiments were not done on the original FNO data and setup, and wondered if the KNO can outperform the FNO on that setup? Clearly, the NS example is not the original autogressive setup from the first 10 time steps to the last 40. The Darcy flow results for FNO is also worse than other what FNO can achieve (implementation updated after the release of the original paper, but the update is from 2 years ago).
- "In section 2.1, integral operators: FNOs use an implicitly-defined, dense, matrix-valued kernel that couples all channels of the architecture. In contrast, the KNO enforces sparsity on this global matrix by utilizing a diagonal matrix-valued kernel."
Can you clarify this? As I understand it, your approach in each kernel integral layer does not use in_channels $\times$ out_channels kernels. Instead, you employ a number of kernels equal to the number of input feature maps/functions, performing integration independently for each channel without interaction across channels. While this indeed reduces the parameter count significantly, it may compromise performance, particularly in comparison to FNO. Moreover, channel interaction is standard in CNNs. Could you provide an explanation or rigorous justification for why this method would not lead to accuracy degradation?

---

### Official Review · Reviewer_jdPE · 2024-11-02

**Soundness:** 4
**Presentation:** 4
**Contribution:** 3
**Rating:** 8
**Confidence:** 4

**Summary:**

This paper proposed a new Neural Operator learning method: Kernel Neural Operator to learning functionals that transform from function to function using parameterized neural networks. The method is based on decomposing the integral operator into a linear combination of simpler but trainable kernel functions and uses sparse kernel mapping. The author uses a Neural Tangent Kernel to justify the kernel choice, which explains the reasons for the good performance.

**Strengths:**

The paper is written very well. The math and narrative is clear. The KNO method proposed in this paper is promising and the paper provides both a theoretical understanding and practical results. The results show a very large advance in accuracy and reduce in parameters, which is great.

**Weaknesses:**

No significant weaknesses are observed. Good job.

**Questions:**

For better presentation and convenience for the further usage and application of this method, I suggest the author include the ablation study of the kernel choice and its impact on the performance.

Also, I am curious about the sampling grid density and its impact on the model training, due to the model approximates the continuous integration into kernelized transformation discretely.

---

### Official Review · Reviewer_duJG · 2024-11-03

**Soundness:** 3
**Presentation:** 3
**Contribution:** 3
**Rating:** 5
**Confidence:** 4

**Summary:**

This paper proposes the kernel neural operator, which uses kernel-based deep integral operators to achieve better performance with fewer parameters compared to other neural operator architectures, such as FNO and DeepONet. The authors compare against several baselines and many numerical examples, including Burgers’ equation, Darcy flow, and Navier-Stokes, to support these findings.

**Strengths:**

The paper is well-written and clear, and the experimental results are thorough. The proposed method seems to provide an improvement over baselines, but most importantly, at the same time reducing computational cost and the number of trainable parameters. In terms of the method, I find the approach to parameterize the matrix-valued kernel as a diagonal matrix to be particularly interesting: performance is not affected because channel mixing can still occur using the pointwise linear layers. I appreciate the care taken to average over different seeds in the experiments. I also appreciate the acknowledgement of limitations on page 2.

**Weaknesses:**

While the experimental results are impressive, the main weakness of the work is that the experimental tasks may be too simple, since most of the proposed methods already achieve < 2% error across most problems. I would be interested in seeing how KNO compares to FNO and alternatives on more challenging problems, such as a turbulent Navier-Stokes for example.

Also, while Table 2 shows an impressive decrease in the number of parameters used for KNO compared to the other models, there have also been some methods proposed previously that seek to address the same shortcomings of FNO/DeepONet that KNO also seeks to solve. For instance, [1] proposes to factorize the Fourier kernels in FNO, which can lead to improvements in performance while reducing parameter count. Similarly, [2] also proposes locally-supported and compact kernels, which have been shown to improve performance on difficult problems with local, high-frequency features, such as turbulent fluid flows. On regular grids, [2] also implements their local kernels using standard convolutional layers for greater efficiency. How does KNO compare with these methods?

The authors also mention the capability of KNO to perform zero-shot super-resolution, but as far as I can see, the only numerical experiments shown are in Figure 7, which includes a small visualization of super-resolution on Darcy flow. I would be curious to see numerical results in how KNO performs compared to FNO (and perhaps the other models) on zero-shot super-resolution.

**Minor notes:**
1. The in-text reference on lines 34-35 may be too long (it includes all the authors).
2. Table 1: do the last two or three rows correspond to irregular domains? The caption and dashed line in the table do not seem to agree.

**References:**
1. “Factorized Fourier Neural Operators” (2021).
2. “Neural Operators with Localized Integral and Differential Kernels” (2024).

**Questions:**

1. What is the use of the kernel interpolant of the input grid $f_{X_T}$ during training? Is the idea to match the input and output grids for greater efficiency? Does this affect performance of the model?
2. What is the intuition for using radial kernels as opposed to, e.g., more general convolutional kernels which can depend on direction as well? Were anisotropic kernels tested as well?

---

### Official Review · Reviewer_pNUd · 2024-11-04

**Soundness:** 3
**Presentation:** 2
**Contribution:** 1
**Rating:** 1
**Confidence:** 4

**Summary:**

The authors present Kernel Neural Operators, which are constructed as mappings in function space, using the convolution integral directly rather than a Fourier space parametrization to contrcut these operators.

Unfortunately this formulation is mathematically exactly the same as the Localized Integral Neural Operators presented [1], which isn't referenced in the manuscript. As the methods are exactly the same, I see very little novelty, and while I do see some interesting new parts such as the choice of different basis  functions, I find the omission

[1] Neural Operators with Localized Integral and Differential Kernels: https://arxiv.org/abs/2402.16845

**Strengths:**

The method works really well, as already demonstrated in [1]. The merit I see in this paper is a different choice of basis functions compared to [1] and a slightly different method for computing the quadrature. Moreover there is an added numerical example.

**Weaknesses:**

The main issue is the fact that this is equivalent with the Localized Integral Neural Operators in [1] and the fact that [1] is not referenced anywhere in the manuscript despite being highly relevant to this work.

Even if [1] is cited, the novelty is marginal and I would also refrain from re-naming an existing method.

Apart from this major flaw, I do see  some issues with only few baselines being used and rather simple examples being used for the experimental evaluation.

**Questions:**

- Have the authors explored more kernel choices than those in the paper?

---

### Note · Authors · 2024-11-23

I have read and agree with the venue's withdrawal policy on behalf of myself and my co-authors.